# An Interactive Live and Online Cooking Program for Children in Vulnerable Families—An Exploratory Study

**DOI:** 10.3390/healthcare10122389

**Published:** 2022-11-28

**Authors:** Jiyoung Park, Sein Hwang, Seolhyang Baek, Gill A. Ten Hoor

**Affiliations:** 1Institute for Health Science Research, College of Nursing, Inje University, Busan 44720, Republic of Korea; 2Department of Social Welfare Administration, College of Health and Welfare, Inje University, Gimhae-si 50834, Republic of Korea; 3Department of Nursing, College of Nursing, WISE Campus, Dongguk University, Dongdaero 123, Gyeongju-si 38066, Republic of Korea; 4Department of Work & Social Psychology, Faculty of Psychology and Neurosciences, Maastricht University, P.O. Box 616, 6200 MD Maastricht, The Netherlands

**Keywords:** cooking, COVID-19, health status disparities, Internet-based intervention

## Abstract

The COVID-19 pandemic has highlighted the importance of technology for communication and social interactions. Especially for children in low-income families—a vulnerable population suffering from health and digital disparities—the situation worsened during the pandemic. Earlier studies in times of COVID-19 suggested that the children in Korea who usually do homework and dine at community childcare centers (CCCCs, free after-school care places) need to learn more about how to eat healthily and how to interact with others using digital technology. Therefore, to reduce these children’s health and digital inequalities, an interactive live and online cooking program was developed and provided to 313 children and 95 staff members at the 29 CCCCs located in the southern provinces in South Korea. The aim of the current study was to explore the experiences of children and staff with the program. After surveying their experiences, a high degree of satisfaction was found (children: 3.60 ± 0.10; staff: 3.63 ± 0.08 points out of 4.00). Aspects that needed improvement in the program were related to (in)experience in online technology, the frequency and timing of the cooking classes, and the communication between the centers and (online) chefs. In addition, in a word cloud analysis, terms such as ‘fun’, ‘delicious’, and ‘want’ were highlighted for children, and terms such as ‘participating’ and ‘preparation’ dominantly appeared for the staff. In the analysis of negative experiences, terms related to environmental factors such as ‘sound’, ‘hear’, and ‘voice’ were highlighted. This novel but preliminary approach for children from low-income families, by integrating cooking with digital technology, indicates that with enough digital support, the CCCCs are a promising platform to promote healthy eating and digital literacy. Optimizing and disseminating these strategies during this pandemic period, and future pandemics, could be beneficial to keep children in their communities healthy, and ultimately reduce socioeconomic health disparities.

## 1. Introduction

### 1.1. Widening Socioeconomic Health Disparities among Children during the COVID-19 Pandemic

The coronavirus disease 2019 (COVID-19) pandemic has affected many families in many ways. For example, people experienced fear, depression, and stress more often [1,2,3], and changed their grocery shopping routines, as well as their food consumption [4,5,6]. Children also experienced similar stress and eating habit changes; prolonged social distancing and continued closure of schools and public facilities because of the COVID-19 pandemic affected children’s sedentary activities and health behaviors. They were often compelled to stay home and were forced to study using digital devices, whilst eating less fresh food and more frozen or convenience food [7]. The changes in their lifestyle ultimately resulted in negative health outcomes. The findings of a survey among 226 children during the COVID-19 pandemic showed that their BMI z-scores, cholesterol levels, triglycerides, and LDL-cholesterol levels were significantly increased, while their vitamin D level was decreased [8].

These health problems resulting from the pandemic related lifestyle changes have been more often reported in low income households. For example, these households have a high tendency to allocate their reduced income or disaster relief funds to medicine expenses or housing rent first, and consequently purchase cheaper junk food or instant food with the remaining money [1,9,10]. Moreover, Caldwell et al. (2021) [11] suggested that these families tend to use food as way of relieving stress and often consume instant or frozen food as a main meal. When these daily life habits are prolonged, the health status of poorer families, including that of children, can easily deteriorate [12,13,14,15], while wealthy households are likely to maintain their daily routine without being affected by the pandemic. Thus, it is likely that household income-related health disparities were worsened by the COVID-19 pandemic [16,17]. Bambra et al. (2020) [18] explained that health inequalities might increase during the COVID-19 pandemic due to the ‘syndemic’ nature of COVID-19, as poor people with chronic diseases were more likely to have fatal type of the disease while suffered from bigger possibilities of losing job or having financial difficulties because of the general lockdown policy and the economic crisis. In addition, Masonbrink and Hurley (2020) [19] mentioned that the long-term school closures have a large impact on children in poverty, who are highly dependent on school-based services for nutritional, physical, and mental health needs. The investigators emphasized the importance of the expansion of vital nutrition programs, including meal distribution, and special supplemental nutrition programs for children, as well as the urgent need to develop various remote and in-person outreach strategies to reach these at-risk children.

### 1.2. The Community Child Care Centers: A ‘Second Home’ in South Korea

In the 1950s, after the Korean War that brought the country extreme poverty and led to mass production of orphanages, a facility called ‘study room’ was created by non-government organizations (NGO) and religious groups in local communities and towns. These study rooms helped poor children across the country, resulting in the current community childcare centers (CCCCs). In 2004, the Korean government amended the ‘Child Welfare Act’ to legislate these CCCCs. They are the recognized child’s welfare institutions in South Korea, which do not only provide after-school care for vulnerable children, but also comprehensive services, including meal services, health care, learning assistance, and counselling [20]. The physical and social environments of CCCCs positively affect children’s eating behaviors and health [21,22,23,24]. Currently, there are more than 4000 CCCCs nationwide, used by more than 100,000 children in low-income families (aged 2–19 years). Therefore, CCCCs are now known as the ‘second home’ for low-income children in South Korea. Although it may be an institution with a public stigma in terms of being a facility only for children in poverty, it is currently the optimal place to provide interventions to children in vulnerable families in South Korea. However, as social distancing and lockdowns policies have without a doubt applied to the second home since COVID-19, the CCCCs have underwent a few limitations such as repeated facilities closures and re-opening, prohibition of eating any food inside, and restriction of number of children and visitors so that these changes have negatively affected the children’s health.

### 1.3. Applying New Information and Communication Technologies (ICT) to Reach At-Risk Populations during the COVID-19 Pandemic

The COVID-19 outbreak and associated social distancing and lockdowns at schools, workplaces, and markets have brought about many challenges in the ‘social connection’ in our daily life [25]. For example, to support children’s learning during the pandemic, most countries, including South Korea, are conducting online educational classes using digital programs, and non-face-to-face video conferencing is more frequently used, rather than face-to-face classes. Furthermore, these challenges strongly required health promotion professionals to change intervention delivery and implementation. New information and communication technology (ICT) (e.g., Smartphone, Zoom, WhatsApp, etc.) provides a convenient and effective solution in this regard [25,26]. During the COVID-19 pandemic, innovative technology enabled the remote maintenance of social connections [27]. Furthermore, it allowed the conveyance of digital interventions by which healthy skills and knowledge can be learned and practiced promoting health behaviors and conditions and alleviate loneliness and distress, due to the COVID-19 isolation [28,29]. For example, Shapira et al. (2021) [30] concluded that a short-term digital group intervention via Zoom, which is a relatively simple and effective technique, already has a positive effect on loneliness and depressive symptoms among the elderly during the COVID-19 pandemic. Esentürk and Yarımkaya (2021) [31] showed that a WhatsApp-based physical activity intervention did increase the physical activity level of children with autism spectrum disorder who stay at home due to the pandemic. Although the evidence is still limited, internet-based interventions that apply new ICT strategies, including WhatsApp and Zoom, can reach at-risk populations effectively and safely during the COVID-19 pandemic era (and beyond).

So far, limited research has focused on non-face-to-face improvement of eating habits for children from low-income families in Korea, except for the study by Lee et al. [32], which examined the use of mobile phone application to provide nutrition education and tailored diet information for the improvement of children’s eating habits. Beyond the attempt of mobile phone intervention, proper utilization of digital instrument(s) to help children learn healthy eating knowledge or practice has seldom found in the country, despite Korean government provided vulnerable children with computer devices and allowed free internet access for educational purposes [33]. According to public reports [34,35], these children are likely to spend after-school hours using the digital devices for entertainment or daily life purposes, such as ordering delivery food, because their parent(s) are away and nobody takes care of them at home. Thus, the application of the new technology “Zoom” to a cooking class that supports healthy eating in children from low-income families may be a substantially meaningful approach for the prevention of intensifying health inequality among children due to the COVID-19 pandemic. Therefore, this study aims to operate a Zoom-based interactive live online cooking program for children using CCCCs, and to investigate the effects and possibilities for future improvement of that program based on the participants’ experiences.

The objectives of the current study were (1) to examine the satisfaction level and experiences of children and staff at CCCCs with the interactive live and online cooking program, and (2) to receive their potential suggestions for program improvement.

## 2. Method

### 2.1. Study Design and Setting

This study is a preliminary pilot study that uses an interactive live and online cooking program and describes the program experiences of children (aged 5–16 years) and staff at CCCCs. This study provides implications for the implementation and dissemination of the interactive live and online cooking program suitable for the COVID-19 pandemic era.

As this investigation adopted community-based participatory research (CBPR) framework which community stakeholders and researchers engage as equal partners, two public health centers were responsible to recruit participants and operate the program. The research team analyzed the data and summarized the points for improvement. In the recruitment process, the public health center confirmed their attendance at the center, and all children who wanted to participate in the program did attend. In two communities, 313 children and 95 staff members from 29 CCCCs participated in this interactive live and online cooking program in August 2021. Zoom was used to teach skills and deliver information to the remote target population [30]. In this study, the technical features and advantages of Zoom were actively utilized. The studio where the chef demonstrated healthy cooking was in the community where 14 of the CCCCs are located; the mean distance from the studio to these CCCCs was about 11 km. In addition, 15 CCCCs in another community (approximately 104 km away from the studio) participated in the program as well (see Figure 1). 

### 2.2. The Interactive Live and Online Cooking Program 

#### 2.2.1. Program Preparation

The motto of this cooking class was “Make Your Own Healthy Meal Using New Information & Communication Technologies (ICT)”. Considering children’s food preferences, Irani sushi with beef topping as the main meal and a strawberry milk drink as the dessert were chosen for the menu [36,37]. Three days before the live and online cooking class, manuals were delivered by e-mail to the 29 participating centers. This manual provided an outline of the program, including the subject and menu of the cooking class, the required time, recipes for cooking the food, and the ingredients provided to the centers. In addition, precautions that must be taken when participating in the program, how to use the Zoom technology, and a connection address were provided in detail. The boxes that contained the ingredients and related items for the cooking class were delivered directly to the centers in the morning of the program operation day (see Appendix A).

#### 2.2.2. Program Operation 

The program consisted of a one-hour live and online cooking program. The cooking classes were held simultaneously at the studio, with a chef and operational staff, and at the centers for children; at least one center and a maximum of four centers participated in the program at the same time. In the studio, one chef and two operational staff members were involved. One staff member assisted the chef in the cooking process, and the other assisted with the technical aspects, such as the Zoom connection and video shooting. At the center, one operational staff member of the research team visited the center an hour before the class started and ascertained the preparation status of the food ingredients, the technical aspects of digital readiness, and the Zoom connection status with the studio. The children were divided into four teams of three children, while all the staff at the centers participated in the cooking class together. In case the center had a large TV monitor, the image was transmitted through the monitor; in other cases, a communication device, such as a laptop or mobile phone, was used instead. The program started with the preparation of the Irani sushi with beef topping, followed by preparing the strawberry milk drink. Because active communication and interaction with the chef on the screen during class was deemed very important, the chef asked the children for physical signals, such as drawing a circle by hand, to make sure that all participants could follow the program at the same speed. In addition, the chef explained the importance of making and eating dishes using healthy ingredients, such as vegetables and fruits. 

After the cooking class, the children and staff filled out a questionnaire. The time taken for the questionnaire was approximately 10 min. In addition, due to the COVID-19 pandemic regulations in South Korea, eating the food inside the center was forbidden. Therefore, each child made his/her own lunch box with the prepared food and took it home.

### 2.3. Instruments

The questionnaire used in this study included items to measure the participants’ general characteristics, program satisfaction, and program experiences.

#### 2.3.1. Program Satisfaction and Further Suggestions 

The questionnaire was developed based on the main components proposed by Larsen et al. (1979) [38] for the development of a general scale for measuring client and patient satisfaction. Its components are as follows: physical surroundings, support staff, kind/type of service, treatment staff, quality of service, amount, length or quantity of service, outcome of service, general satisfaction, and procedures. Based on these components, a tool was developed, which fitted within the boundaries of the program circumstances. The tool consisted of 11 questions on a 4-point Likert scale, with a higher score reflecting higher satisfaction with the program. The content of the satisfaction questionnaire for children and staff at CCCCs was the same. If they were satisfied with this program, they were asked to provide reasons for their satisfaction. In addition, based on the above components, they were asked to suggest points of improvement. Finally, they could describe impressions and suggestions using their own words related to improvements of the interactive live and online cooking program. All questionnaires were in Korean.

#### 2.3.2. Cooking Program Experiences 

Participants were asked if they had participated in online cooking classes in the past and were also asked about the need for a program to prepare their own food and the need for a program using online technology.

### 2.4. Data Collection

The ethics review board of the researcher’s institution approved this study (IRB No. 2022-02-015), and all procedures complied with the institution’s ethics policy. The children and staff who participated in the program filled out the questionnaire immediately after the cooking program ended. The assistant of the research team provided the participants with the questionnaire and asked them to answer honestly about their feelings on participating in the program. 

### 2.5. Data Analysis 

This study is a preliminary study that was performed to understand the participants’ experiences, including program satisfaction and suggestions, of the cooking program. To solve the research questions systematically, the data of the survey were quantitatively analyzed, and simultaneously, the opinions of the participants from the survey were analyzed qualitatively. For the quantitative approach, descriptive statistics, including the mean and standard deviation and percentage, were applied to analyze the data, using the Statistical Package for the Social Sciences (SPSS) version 21.0. In addition, qualitative data from a total of 202 participants (167 children; 35 workers) who had described their participation experiences via open-ended responses were analyzed. First, descriptions by the respondents were translated from Korean to English, incorporating a conversational style as much as possible to vividly convey their experiences. The translated data were then uploaded onto NVivo 12, a qualitative analysis software, and analyzed and expressed with word clouds. It can be quite challenging for researchers to utilize computer-assisted qualitative data analysis software (CAQDAS), including NVivo 12. However, these programs increase the efficiency of data management and transparency in analysis. In particular, the ‘word clouds technique’ utilized in this study helps to create a comprehensive description of the phenomenon through the visualization of research results [39]. 

## 3. Results

### 3.1. Sample Characteristics

Table 1 shows the general characteristics of 313 children and 95 staff who participated in the interactive live and online cooking program.

### 3.2. Program Satisfaction

The overall program satisfaction score was 3.62 (±0.46) (out of 4) points for the children and 3.64 (±0.45) points for the staff, showing generally high satisfaction (see Table 2). Both the children and the staff indicated overall satisfaction across all five sub-domains, and the staff showed a high level of satisfaction with the food ingredients, main menu, and dessert menu used in the program. 

In response to the question that asks about the reasons for being satisfied with the cooking class, 45.1% of the children and 62.0% of the staff answered that ‘It was fun to cook together using the zoom program’. In addition, 23.6% of the children and 31.5% of the staff answered that ‘The program was convenient because it was an online program that doesn’t require any travel away from the center’. Moreover, 29.0% of the children and 21.7% of the staff responded that ‘It was such a novel program that I have never experienced before’. It should also be noted that the statement ‘Seeing other people having a lot of fun while cooking’ was also listed as a reason for program satisfaction by 12.8% of the children and 40.2% of the staff. Finally, 36.4% of the children and 34.8% of the staff answered that ‘I was able to learn how to make simple and healthy food through this program’.

Among the very few participants who expressed dissatisfaction with the program, some participants selected ‘I am not familiar with using online technology (e.g., ZOOM program)’ (two children and one staff member) and ‘The cooking space was too small’ (three children and two staff members) as reasons for their dissatisfaction. In addition, eight children responded that their dissatisfaction was related to their preference and taste of the food.

Suggestions made by the program participants are shown in Table 3. First, 20.8% of the children and 16.8% of the staff suggested that ‘CCCCs needs opportunities to use digital technology’, and 14.7% of the children and 35.8% of the staff suggested that ‘a high enough number of cooking programs should be provided (not just once)’. Moreover, suggestions such as ‘Ensuring enough time for the cooking program’, and ‘Improving online communication and interactions between CCCCs and program operators’ has been mentioned a lot (see Table 3).

Lastly, as displayed in the word clouds, ‘Fun’ stands out as the major keyword in the experiences of the children, as well as the staff. The children responded, for example, with the following statements: “It was helpful to learn how to cook. It was a big joy and interesting to make food with my friends.”, “Given the COVID-19 situation, I liked how we did an interactive-live online class without meeting people face-to-face. I would like to do it again.” The staff’s responses included the following statements: “I think it gave the kids a good experience of cooking in an innovative way.”, “Due to the limited space, it was difficult for many children to participate in the cooking class. No matter how good the content of interactive live and online cooking program is, it is difficult to proceed with the class smoothly if the class is not thoroughly prepared due to lack of manpower and lack of time. Since it was the first time for us to have an interactive live and online class, we were not ready enough for the class. Nevertheless, the children were able to happily participate and thank to the program staff”.

To summarize the narrative responses, the participants ‘enjoyed making food together and wished for this kind of opportunity again because the food was great’. The staff reacted positively about the children ‘having fun and participating actively’, but emphasized the need for more preparation by both CCCCs and program operators in terms of cooking supplies, tools, online education equipment, etc. They also pointed out the program was difficult because it was their first time engaging with an online educational program and that the low volume and poor audio quality need to be improved (see Figure 2).

## 4. Discussion

As the COVID-19 pandemic progressed, the socioeconomic health disparities among children became larger and were accelerated. Therefore, the development of digital technology-based health promotion strategies that can be adapted to the circumstances during the COVID-19 pandemic also needed to be accelerated. As a result of this interactive live and online cooking program using Zoom at CCCCs for children from vulnerable families during the COVID-19 pandemic, a high level of satisfaction for the program was reported; however, several elements need to be improved, as the participants’ experiences revealed. Cooking and nutrition programs for children are effective for cultivating healthy eating behaviors in children and the related social determinants, including knowledge and self-efficacy regarding cooking [40,41]. However, such programs are rarely implemented under the current conditions, where there is no support for professional instructors nor a cooking environment [42]. Given the situation, this study conducted an interactive live and online program, led by only one professional instructor, for many children at different CCCCs. The program also involved food that does not require a separate cooking environment. Therefore, the program in this study was implemented in ways that resolve the limitations of cooking programs that have been available until now, and the program was found to be highly satisfying for both children and staff. Specifically, the participants were satisfied with ‘the fun of the cooking class itself, the real-time exchange with multiple people and convenience from using the Zoom program, and the novelty of digital experience’. The program considered a way to make the program applicable at home by selecting food/dishes that children can easily cook and do not require a separate cooking space. This aspect will allow both CCCCs and households to improve the dietary care and each other’s relationship by the digital technology [32]. To maximize these expected effects, it is necessary to expand the program in terms of frequency and target participants in the future by reflecting ‘the need for having more cooking programs that use digital technology’, as expressed by the participants.

Despite the high satisfaction with the program, there were several points for improvement; A few children answered that they had not been satisfied when cooking foods that they did not like before, while the staff stated that lack of skills for the digital programs and poor preparation of food ingredients made them not satisfied. In relation to the children’s negative responses to unfamiliar food, a previous study about a community meal program for the elderly in Korea suggests that it is also difficult for the elderly to understand the recipes of unfamiliar foods and that familiar ingredients and recipes reduce the resistance of program participants [43,44,45]. Pertaining to the problems raised by the staff, Barbosa and Barbosa (2019) [46] suggest improving the effectiveness of online education by offering ‘practice’ classes to show the participants how to use technical tools, ensuring that the participants are in a quiet environment without distractions, have a high-quality computer, and use audio equipment or headphones with a microphone when participating in class. Some CCCCs where the program was implemented had inadequate equipment or infrastructure for effective digital communication, although all CCCCs should be equipped with these tools to communicate digitally with families and related public institutions [35]. The researchers checked these aspects before implementing the program, but these checks were apparently insufficient. Therefore, it will be necessary in the future to perform concrete checks (rehearsals or tests) and provide support (microphone, headsets, etc.). In addition, it is necessary to allocate sufficient time for the preparation/organization of the cooking program and to prepare a detailed checklist related to the preparation needed to solve the problems pointed out through the active communication between CCCCs and the program operators. In addressing barriers such as the lack of cooking tools and online educational equipment and shortage of trained professional workforce in the future, education and welfare administrative authorities should recognize the educational value of cooking activities in general, which is not limited to the program presented in this study, and provide appropriate support for facilities, in addition to equipment [47].

During this pandemic era, the online learning became a main educational way between teachers and children, however, it was mostly implemented in a unidirectional manner that did not fully allow children to communicate in real time. It is well known that effective learning is optimized by using two-way communication [48,49,50]. Handgraaf et al. (2012) [51] asserted that, when sufficient interaction is guaranteed, a digital classroom is by no means inferior to a face-to-face classroom and can adequately function as a substitute for face-to-face communication. Hu et al. (2021) [48] showed the need for more interactive online teaching preparation to address young children’s learning needs. Therefore, the cooking program in this study intended to promote multi-directional communication between program operators and participants at the respective CCCCs, ensuring an interactive live environment. This arrangement is one of the reasons why words as ‘fun’ and ‘participation’ appeared in the word clouds. A program that allows a child to communicate with teacher and have a fun during digital learning- so called Edu-entertainment- is more likely to motivate the child to change his/her habit or lifestyle in healthy way [52]. Even though various teachers have received professional training on remote instruction, this was not sufficient [53]. Therefore, instructors in charge of digital education should implement digital classes, taking into consideration digital technology, with concrete knowledge of ways to promote interactions between instructors and participants [54].

The younger generation is likely to form healthy eating habits as they have more opportunities to cook for themselves [55]. During the COVID-19 pandemic, the tendency to eat out decreased around the world, and the frequency of eating at home is rapidly increasing [56]. However, in this pandemic situation, unhealthy eating habits, such as eating food to relieve loneliness or stress and eating fast food or frozen food instead of fresh vegetables and fruits, are likely to be reinforced [57,58]. Therefore, it is important to form and maintain healthy eating habits during this pandemic. In conclusion, considering that CCCCs are partially in charge of food for children in low-income families, CCCCs should expand their role beyond simply providing meals and snacks to a place where children can experience nutritional education. To fulfill this role, CCCCs should have no restrictions in using digital programs that can be applied even in crises such as the COVID-19 pandemic. Moreover, various cooking programs should be developed with the aim of improving digital literacy of both the service providers that offer the cooking classes using digital programs and the users of this service.

### Limitations and Further Suggestions

This study is a first attempt to examine user satisfaction and improvement points of an interactive live and online cooking program that enrolled 408 participants in CCCCs in Korea. The CCCCs involved in this study included 29 centers located in 2 cities in the southern part of Korea. Therefore, expectations about the effectiveness of the program were preliminary. A well-designed experimental field-study with a comparison group is recommended to verify the effectiveness of the program, not only in terms of lifestyle changes (e.g., dietary behaviors), but also in terms of physiological indicators (e.g., blood pressure and body mass index). Additionally, future studies should examine the influence of other factors, such as current cooking behaviors, family income, age, and regional distributions, to better inform governmental decision making.

Many program participants in this study responded that in addition to improving their digital literacy, the expertise of the program operators should be improved. In this study, one operational staff member monitored the technical aspects of the digital apparatus before the program, and at least one or two assistants supported the participants during the entire period of the program. To ensure the service quality, we measured their satisfaction with the questionnaire item ‘Service provided by Staff’ in Table 2. Both the children and CCCC staff rated 3.64~3.71 out of 4, meaning ‘highly satisfied’. An accompanying guardian, including staff and parents, can increase the efficiency and effectiveness of the program, especially for children. This will vary depending on the child’s age and technology competency regarding digital applications. However, since the purpose of this study was not to verify this, follow-up research on effective program operation standards using digital devices is needed in the future.

Lastly, the centers used to directly ‘provide’ children with meals and snacks rather than ‘shape’ life-long eating behaviors, including teaching the children how to cook, and how to eat in a healthy manner, etc. As result of the simple provision of foods and without behavioral intervention, the children at the CCCCs became just eaters and had no chance to improve their eating habits. As the COVID-19 pandemic has changed this world to adapt digitally convergent lifestyle, it is the right time to reform the environment of CCCCs in that way. In addition, the CCCCs has now faced to change their classical role as a place for foods for the vulnerable children to digital platform for healthy nutrition.

## 5. Conclusions

Most of the participants expressed very high satisfaction with the interactive live and online cooking program, which can be attributed to program elements such as direct participation in making the food, live interaction with multiple people and the convenience of the Zoom program. Notwithstanding these positive elements, it was found that digital education/experience programs for CCCCs should not be implemented as a one-time event, but must be planned, improved, and implemented regularly to achieve a greater impact.

In conclusion, CCCCs are a significant place (i.e., a second home) for children with vulnerable backgrounds in South Korea, but careful attention is needed at the community and policy level to ensure that CCCCs can continuously expand their role as a digital learning platform that can provide children with experiential dietary education, shifting away from the role of simply ‘providing’ care and meals to children from low-income families.

## Figures and Tables

**Figure 1 healthcare-10-02389-f001:**
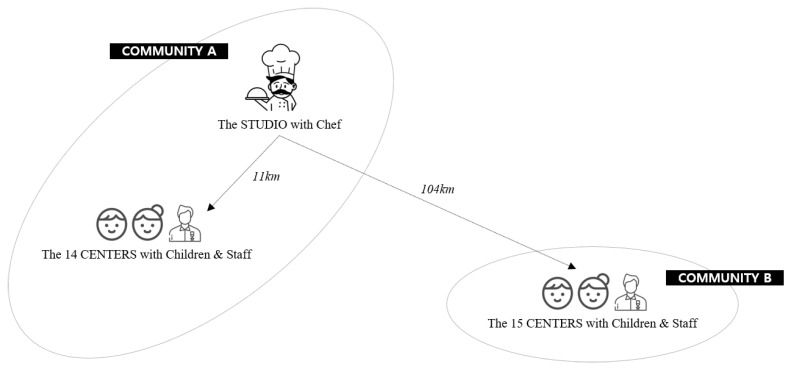
Location and setting for the interactive live and online cooking program.

**Figure 2 healthcare-10-02389-f002:**
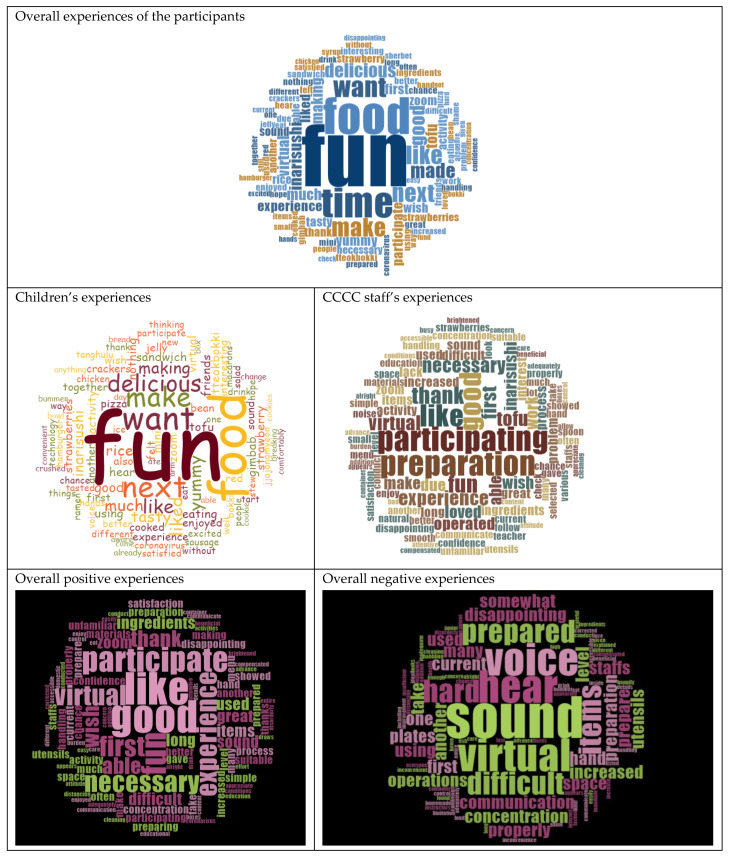
Word cloud analysis of the participants’ impressions and suggestions.

**Table 1 healthcare-10-02389-t001:** Demographic profile of the participants.

Characteristics		Categories	N (%)
Children (*n* = 313)	Gender	Boy	156 (50.2)
		Girl	155 (49.8)
	School level	Primary	291 (94.2)
		Secondary	15 (4.9)
		High school	2 (0.7)
		Kindergarten	1 (0.3)
	Years of CCCC attendance	Less than 1 year	64 (24.0)
		1~3 year	92 (34.5)
		3–5 year	69 (25.8)
		Longer than 5 years	42 (15.7)
	Previous cooking experience using digital program	No	171 (55.5)
		Yes	137 (44.5)
CCCC staff (*n* = 95)	Gender	Man	17 (17.9)
		Woman	78 (82.1)
	Age	<20 years	4 (4.4)
		Twenties	27 (29.4)
		Thirties	15 (16.3)
		Forties	19 (20.7)
		Fifties	22 (23.9)
		Sixties	5 (5.4)
	Job title	Chief of CCCC	22 (24.2)
		Social worker (general help)	45 (49.5)
		Cook (makes lunches and dinner)	14 (15.4)
		Other (e.g., volunteers, support staff)	10 (11.0)
	Dates of employment at CCCCs (year)	Less than 1 year	34 (38.6)
		1–3 years	14 (15.9)
		3–5 years	11 (12.5)
		5–10 years	14 (15.9)
		Longer than 10 years	15 (17.1)
	Previous cooking experience using digital program	No	80 (85.1)
		Yes	14 (14.9)

**Table 2 healthcare-10-02389-t002:** Participant satisfaction with the program.

Item	Response *			
	Children (*n* = 313)		CCCC Staff (*n* = 95)	
	Mean	*SD*	Mean	*SD*
Physical surroundings				
I am satisfied with the Zoom technology used by the interactive live and online cooking program	3.45	0.75	3.53	0.64
Type of program				
I am satisfied with the food ingredients (e.g., tofu) used in the interactive live and online cooking program	3.65	0.64	3.74	0.51
I am satisfied with the main menu (e.g., the inari sushi) of the interactive live and online cooking program	3.62	0.71	3.76	0.50
I am satisfied with the dessert menu (e.g., strawberry milk drink) of the interactive live and online cooking program	3.65	0.73	3.70	0.51
Quality/outcome of program				
I am satisfied with the quality of the interactive live and online cooking program	3.65	0.55	3.56	0.60
I think that the interactive live and online cooking program helped to improve the participant’s healthy eating habits	3.62	0.55	3.55	0.64
Service provided by staff				
I am satisfied with the competence of the staff who carried out the interactive live and online cooking program	3.66	0.59	3.64	0.55
I was comfortable with the staff members during the interactive live and online cooking program	3.71	0.53	3.66	0.48
General satisfaction				
I am satisfied with the overall service of the interactive live and online cooking program	3.68	0.60	3.56	0.58
I intend to take part in an interactive live and online cooking program again if necessary	3.65	0.65	3.64	0.61
I intend to recommend the interactive live and online cooking program to my friend(s) or family	3.37	0.85	3.67	0.60
Overall score	3.62	0.46	3.64	0.45

****** 1 = strongly agree, 2 = slightly agree, 3 = slightly disagree; 4 = strongly disagree*.

**Table 3 healthcare-10-02389-t003:** Participants’ suggestions for improving the program.

Item	N (%)	
	Children (*n* = 313)	CCC Staff (*n* = 95)
The CCCCs need opportunities to utilize digital technology.	65 (20.8)	16 (16.8)
There should be a higher number of cooking programs.	46 (14.7)	34 (35.8)
Online communication and interaction between CCCCs and operational staff need to be improved.	33 (10.6)	10 (10.5)
Enough time for implementing the cooking program should be guaranteed.	37 (11.9)	9 (9.5)
Program operation skills of staff members need to be improved.	20 (6.4)	2 (2.1)
More staff or assistants is needed to support the program.	13 (4.2)	12 (12.6)
Enough food ingredients should be available for the CCCCs.	17 (5.5)	5 (5.3)
The food menus for healthy eating need to be improved.	23 (7.4)	2 (2.1)
The expertise of operational staff needs to be improved.	12 (3.9)	3 (3.2)

## Data Availability

The data presented in this study are available on request from the corresponding author. The data are not publicly available due to the ownership of data.

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
