# Peer review of "An Interactive Live and Online Cooking Program for Children in Vulnerable Families—An Exploratory Study"

_healthcare, 2022, doi:10.3390/healthcare10122389_

Round 1

Reviewer 1 Report

1.     How to ensure that students use digital devices effectively to watch programs instead of other content? Does the student need to be accompanied by a parent or adult?

2.     I suggest that the manuscript should include the measurement and recording of physiological indicators of students and staff during the process of the project, including blood pressure, body temperature, heart rate, etc. In this way, the intervention effect of the program on students' health can be fully illustrated.

3.     This study lacks control group, and the results are uncertain. According to the experimental setup of this study, there should be at least a comparative analysis between student groups watching and not watching the program, participating in the experience and not participating in the experience.

4.     Please explain in detail how this study can contribute to the development of student healthcare intervention policies.

5.     Last but not least, in my opinion, the main reason why children from poor families don't eat healthy is the lack of healthy food, not cooking shows.

Author Response

thank you for your helpful feedback. Please find our response attached

Reviewer 2 Report

The study titled An Interactive Live & Online Cooking Program for Children in Vulnerable Family during the COVID-19 Pandemic in South Korea: Exploring the Experiences of Children and Staff at the Community Child Care Centres was reviewed.

I'm in the opinion that overall manuscript is sound for publication after some revisions. Please find some suggestions in the attached file.

Author Response

(The authors gave the same response as above.)

Reviewer 3 Report

The topic is very interesting and innovative. The manuscript is well prepared and easy to follow. 

Besides the word clouds presentation of results is timely and  engaging.

However, I have few minor comments to revise and correct

1- objective should be split into 2 objectives or into primary and secondary objective

2- Reconsider writing style of the research questions, that should be mentioned in a scientific way

3- age group of children should be mentioned

4- Line 170-2: what is the reference of this information

5- Tables used should be all unified in format, space, etc.

   5.1- In tables: specify what is "less than1, 1 ~ 3, 3-5, etc." in children and staff'

  5.2- In tables: specify what is " chief. social worker, cook, etc.

6- Conclusion is very long and contains some implications. Therefore, these implications should be moved up to the section 4.1 (Further suggestions"

Author Response

Thank you for your kind feedback. Based on your feedback, we made the following changes:

  • The objective and research questions have been revised as suggested by the reviewer (point 1 and 2):

“The objectives of the current study were 1) to examine the satisfaction level and experienc-es of children and CCCCs staff with the interactive live&online cooking program, and 2) to receive their potential suggestions for program improvement.”   

  • We added the age-range for children in section 1.2 (line 82) and section 2.1 (line 144).
  • Tables are all in the same format now, additional information is given where information was unclear
  • Conclusion is shortened and implications are moved to the discussion (section 4.1. Further suggestions). The conclusion now reads:

“Most of the participants expressed very high satisfaction for an interactive live & online cooking program, which can be attributed to program elements such as direct participation of making food, and live interaction with multiple people and convenience through ZOOM program. Notwithstanding these positive elements, it was found that digital education/experience programs for CCCCs should not be implemented as a one-time event but must to be planned, improved and implemented regularly for more effect.

In conclusion, the CCCC is a significant place (i.e., second home) for children with vulnerable backgrounds in South Korea, but careful attention is called for at the community and policy level to ensure that CCCCs may continuously expand their role as a digital learning platform capable of providing children experiential dietary education, shifting away from the service of simply ‘providing’ care and meals to children from low-income families.”

Reviewer 4 Report

Overall a good manuscript looks at interactive training for kids to learn better-eating habits. The methodology has been clearly defined.

The questionnaire used to assess the participant satisfaction with the program by both kids and staff is very appropriate. I like that the researchers also had a feedback questionnaire to improve it. 

The word clouds display also is innovative and shows the main positive and negative feelings associated with the training program. Overall it would be an excellent study to base future training programs for kids online. 

English and grammar are acceptable, and it requires spell check. 

Author Response

thank you for your kind feedback. Language is checked once more.

Round 2

Reviewer 1 Report

The author has made a good answer and revision to the review comments. Television communication has a great impact on people's thinking and behavior, but more research is needed to really improve the level of healthy eating among poor children. The author's research provides some professional basis for the government to make decisions.

In the context of the COVID-19 pandemic, the authors need to classify and study the effects of TV communication on the healthy diet of poor children according to the family income, age and regional distribution of the population, which will provide a more favorable scientific basis for government decision-making. At the same time, the authors should track and monitor the physical health indicators of all participants under the experimental conditions. These physiological indicators can be monitored and measured remotely by electronic instruments.

Therefore, the content and method of this study need to be further improved. First, the author needs to conduct an in-depth investigation into the cooking behavior changes of poor children's families after the end of the program. Secondly, after the end of the program, the changes of children's physical health index should be tracked for one month. Finally, the authors also need to accurately investigate the source of food in poor children's homes, including before, during and after the program.

Author Response

Thank you for the compliments and suggestions. As the study was already conducted, we are unable to change either our research question or methodology. therefore we added the suggestion by the reviewer as a future suggestion: 

A well-designed experimental field-study with a comparision group is recommended to verify the effectiveness of the program, not only in terms of lifestyle changes (e.g. dietary behaviors), but also in terms of physiological indicators (e.g. blood pressure and body mass index). Additionally, future studies should examine the influence of other factors such as (for example) current cooking behaviors, family income, age, and regional distributions to better inform governmental decision making.

Reviewer 2 Report

-

Author Response

This reviewer did not give any further suggestions to improve our manuscript. However, we did proofread our paper one more time, made some final language corrections, and improved the flow where needed.